# PI3K/AKT Signaling Tips the Balance of Cytoskeletal Forces for Cancer Progression

**DOI:** 10.3390/cancers14071652

**Published:** 2022-03-24

**Authors:** Shuo Deng, Hin Chong Leong, Arpita Datta, Vennila Gopal, Alan Prem Kumar, Celestial T. Yap

**Affiliations:** 1Department of Physiology, Yong Loo Lin School of Medicine, National University of Singapore, Singapore 117593, Singapore; phsdes@nus.edu.sg (S.D.); vennila.gopal@u.nus.edu (V.G.); 2NUS Centre for Cancer Research (N2CR), Yong Loo Lin School of Medicine, National University of Singapore, Singapore 117597, Singapore; leonghc@u.nus.edu; 3Cancer Science Institute of Singapore, National University of Singapore, Singapore 117599, Singapore; a.dattacsi@nus.edu.sg; 4Departments of Pharmacology, Yong Loo Lin School of Medicine, National University of Singapore, Singapore 117600, Singapore; 5National University Cancer Institute, National University Health System, Singapore 119074, Singapore

**Keywords:** PI3K/AKT, cytoskeleton, cancer, chemotherapy, clinical trial

## Abstract

**Simple Summary:**

The PI3K/AKT signaling pathway plays critical roles in regulating a series of cellular changes to promote tumor development and progression. The cytoskeletal network, comprising of the microfilaments, microtubules, and intermediate filaments, is known to be regulated by signaling cascades, which lead to dissemination of primary tumors and thus worsen clinical outcomes. Both aberrant activation of the PI3K/AKT pathway and alteration of cytoskeletal structures are highly prevalent in cancer cells. However, it is not fully understood how the crosstalk and feedback between PI3K and the cytoskeleton could cooperatively lead to cancer progression and a poorer patient prognosis. Herein, we discuss the molecular and cellular regulation between cytoskeletal proteins and the PI3K/AKT signaling pathway, and how these two orchestrate a regulatory process that aids cancer progression. Our review also summarizes recent advances in the clinical development of PI3K and cytoskeleton targeting agents, thereby providing insight into the development of novel therapeutic approaches targeting the interplay between PI3K and the cytoskeleton for cancer management.

**Abstract:**

The PI3K/AKT signaling pathway plays essential roles in multiple cellular processes, which include cell growth, survival, metabolism, and motility. In response to internal and external stimuli, the PI3K/AKT signaling pathway co-opts other signaling pathways, cellular components, and cytoskeletal proteins to reshape individual cells. The cytoskeletal network comprises three main components, which are namely the microfilaments, microtubules, and intermediate filaments. Collectively, they are essential for many fundamental structures and cellular processes. In cancer, aberrant activation of the PI3K/AKT signaling cascade and alteration of cytoskeletal structures have been observed to be highly prevalent, and eventually contribute to many cancer hallmarks. Due to their critical roles in tumor progression, pharmacological agents targeting PI3K/AKT, along with cytoskeletal components, have been developed for better intervention strategies against cancer. In our review, we first discuss existing evidence in-depth and then build on recent advances to propose new directions for therapeutic intervention.

## 1. Introduction

### 1.1. PI3K/AKT Signaling Pathway in Cancer

The phosphoinositide 3-kinase (PI3K)/AKT pathway is a vital oncogenic pathway that plays critical roles in multiple aspects of cancer hallmarks, including cell survival, metabolism, metastasis, and angiogenesis [1,2,3,4]. There are three classes of PI3Ks, stratified by sequence homology and substrate preference. The PI3Ks function to generate specific phosphoinositides inside cells, where class I PI3Ks synthesize phosphatidylinositol 3,4,5-trisphosphate (PtdIns(3,4,5)P_3_ or PIP_3_), class II PI3Ks produce phosphatidylinositol 3-phosphate (PtdIns3P) and phosphatidylinositol 3,4-bisphosphate (PtdIns(4,5)P_2_), and class III PI3Ks generate PtdIns3P [5,6]. All classes of PI3K are key players in mediating multiple cellular processes via the regulation of specific phosphoinositides with district cellular localization [7]. Class I PI3Ks, which are formed as heterodimers of a catalytic subunit (p110α, p110β, p110γ, or p110δ) and a regulatory subunit (p85α (or its splice variants p55α and p50α), p85β, p55γ, p101, or p84), are the most studies class, with extensive understanding of their oncogenic properties (the review focuses on class I PI3Ks herein) [7,8]. PI3Ks are triggered downstream through the activation of receptors, including tyrosine kinase receptors (RTKs), cytokine receptors, and G proteins coupled receptors (GPCRs). The PI3Ks are then recruited to the plasma membrane, catalyzing the phosphorylation of the 3′-hydroxyl group of phosphatidylinositol 4,5-bisphosphate (PtdIns(4,5)P_2_ or PIP_2_) to produce the second messenger molecule PIP_3_ (Figure 1). This lipid conversion process is reversed by PTEN, the antagonist of the PI3K pathway, through dephosphorylation of PIP_3_. As second messengers, PIP_3_ accumulate at the plasma membrane to recruit downstream effector proteins containing pleckstrin homology (PH) domains, to interact with this lipid [1]. One of these recruited proteins is the serine-threonine kinase AKT, which is phosphorylated by phosphoinositide-dependent protein kinase 1 (PDK1). Activated AKT signals to multiple downstream effectors to control diverse cellular functions and determine the cell fate. For instance, mTOR is activated to initiate protein synthesis via S6K and 4EBP [9,10], and inhibit the forkhead box O (FOXO) family transcription factors to promote cell survival and metabolic reprogramming [11,12]. Moreover, the PI3K/AKT pathway has been found to crosstalk with multiple signaling cascades, including the ERK/MAPK, JAK/STAT, and RAS/RAC pathways [13,14,15,16,17]. The extensive crosstalk among PI3K/AKT and other molecules thus forms highly interdependent signaling networks to cooperatively regulate multiple cellular functions of cancer cells and direct the disease progression by enhancing cell proliferation, migration/invasion, and treatment resistance.

As one of the most frequently mutated oncogenic pathways, PI3K/AKT signaling has been identified to exhibit a broad mutational spectrum on various components, leading to hyperactivation of this pathway to promote tumor development and disease progression [17,18,19,20,21,22]. Mutations have been identified on all catalytic subunits (p110α, β, γ, and δ) [23,24,25,26]. Among these subunits, activating mutations on p110α are found in approximately 20–40% of solid tumors, with more than 25 mutation sites identified in colon, breast, and gynecological cancers [27,28,29,30]. These mutations usually contribute to hyperactivation of the PI3K pathway, which leads to oncogenic transformation of normal cells, somatic tumor formation in mice, and increased cancer cell invasion and drug resistance [31,32,33]. Another frequent mutation causing PI3K hyperactivation is the loss-of-function mutation occurring in PTEN, the antagonist of PI3K pathway. About 60–80% of patients with PTEN hamartoma tumor syndromes (PHTSs) carry germline mutations of PTEN, predisposing the patients to increased tumor risk [34,35]. Somatic inactivation of PTEN is also prevalent in a wide range of sporadic tumors, including colon, melanoma, prostate, and endometrial cancers [36,37]. Moreover, expression of PTEN can be inactivated through multiple post-transcriptional and post-translational regulation, which could explain the PTEN inactivation in cancer cells without a germline PTEN mutation [38]. These mutations, occurring to different components of PI3K signaling, can synergize or act independently to induce constitutive activation of PI3K and downstream effectors, which further induces cellular changes to enhance the tumorigenesis and aggression of an established cancer. For instance, remodeling of the cytoskeleton is promoted under PI3K activation, which enhances the metastatic potential of cancer cells. With accumulating evidence from in vitro and in vivo studies to support the mutated molecules of PI3K as drivers of oncogenic transformation and therapeutic resistance, PI3K might be an ideal target for developing anticancer drugs.

### 1.2. Critical Roles of Cytoskeleton in Cancer

The cytoskeleton landscape is a complex dynamic network of filamentous proteins that provides shape and support to the cell, facilitating the transport of molecules, cell division, cell invasion, and cell signaling. The cytoskeleton comprises three components: actin filaments (microfilaments), microtubules, and intermediate filaments (Figure 2). All three filament systems are highly dynamic, altering their organization in response to the needs of the cell. Actin exists in two forms: monomeric globular (G-actin) and polymeric filament (F-actin). These two are under constant dynamic conversion, where G-actin polymerizes into F-actin and adds to the existing filament from its plus end, while F-actin hydrolyses and depolymerizes from its minus end [39]. The balance of the two is controlled by a large group of actin-binding proteins (ABPs) inside cells. ABPs also control the spatial distribution and remodel actin organization in response to signals, leading to the execution of multiple cellular processes like vesicular/protein trafficking, cell migration, and maintenance of cell junctions/polarities [40]. Microtubules (MT) are formed by polymerization of tubulin dimers, consisting either of α- or β-tubulin. Like actin filaments, microtubules have two distinct ends, with a plus end oriented towards the cell periphery and a minus end anchored at the microtubule-organizing center adjacent to the nucleus. MT-binding proteins (MTBPs) regulate microtubule assembly, depolymerization, stabilization, and cross-linkage, where these dynamics of microtubules are vital to cell division, intracellular trafficking, cell growth, and cell death [41]. Unlike actin filaments and microtubules, which are polymers of single types of proteins, intermediate filaments are made up of a number of proteins, with distinct functions [42]. Intermediate filaments (IFs) provide structural support for cells, contributing to cell shape maintenance, cell migration, and cell adhesion [43]. Several members of IFs are also key mediators of the process of transducing external mechanical stresses into cells. For instance, keratins form a network in airway epithelial cells that protects cells against shear stress [44].

Research has shown that the cytoskeleton plays vital roles in multiple stages of cancer progression, with well-illustrated roles in regulating epithelial-mesenchymal transition (EMT) and metastasis [45,46,47]. All three cytoskeletal components collaborate and function collectively to control each step of cell migration, including polarization, formation of protrusions, adhesion, contractility, and transmission. The cytoskeleton spans the cytoplasm and connects the cell nucleus with the extracellular matrix (ECM) to provide the mechanical strength and structural basis for cell movement. During the EMT process, the cytoskeleton is restructured under a coordinated regulation of ABPs, MTBPs, and regulators of IFs, to weaken cell-cell attachment and strengthen cell-matrix adhesions that transform cancer cells from the stationary epithelial type into migratory mesenchymal type [47,48]. These cytoskeletal components further coordinate in a systemic manner that leads to the formation of cellular protrusions like lamellipodia, filopodia, and invadopodia, which ultimately leads to cancer metastasis [48,49,50,51]. In fact, many of the cytoskeletal molecules, such as vimentin and keratin, are established as biomarkers for monitoring EMT and metastasis progress [52]. Besides acting as metastatic factors, the cytoskeleton and its associated proteins are also critical regulators of cancer cell survival and development of multidrug resistance. For instance, actin filaments and ABPs are key regulators of apoptosis, cell growth/proliferation, and angiogenesis [53,54,55,56]. Remodeling of the actin cytoskeleton and altered expression of ABPs, such as tropomyosin, gelsolin, and cofilin, are frequently observed in solid tumors, allowing tumor cells to evade apoptosis signaling, stimulate cell proliferation, and acquire chemotherapy resistance [56,57,58,59,60]. Similarly, studies have also shown that altered microtubule dynamics, with tubulin mutations and differential expression of the isoforms, and MT-associated proteins are critical for developing resistance to chemotherapeutic regimes, including microtubule-trageting agents [61,62,63,64,65]. A vast majority of the cytoskeleton-associated proteins are under concerted regulation of signaling networks, such as the PI3K/AKT pathway, as a response to external and internal stimuli. In addition, changes in the cytoskeleton also feedback to the upstream signaling pathway, which cooperatively reshapes cancer cell behaviors.

## 2. PI3K/AKT in Regulating Multiple Aspects of Cytoskeleton in Cancer Biology

### 2.1. PI3K in Regulating the Actin Cytoskeleton

The activation of PI3K signaling not only mediates many critical cellular functions but also greatly influences cytoskeletal changes [21,66]. The lipid products of PI3K (PIP2 and PIP3) are capable of binding to downstream protein targets containing the PH—which include the guanine nucleotide exchange factors (GEFs) (Figure 3) [67,68]. This, in turn, promotes the activation of the Rho family of small GTPases such as Rac1, RhoA, and Cdc42—which modulate the dynamics of the actin cytoskeleton [69,70,71]. These Rho GTPases drive the polymerization of actin monomers by activating actin nucleators such as the WASP/WAVE proteins via the Arp2/3 complex [72,73]. On activation, the Arp2/3 complex orchestrates the formation of nascent actin filaments by branching off pre-existing actin filaments. Furthermore, Rho GTPases are also known to inactivate cofilin (which is an actin filament-severing and depolymerizing factor) with the help of the LIM kinases (LIMKs) to stabilize actin filaments [74]. The polymerization of the actin monomers at the plasma membrane at the leading edge of migrating cells induces the formation of protrusions such as the lamellipodia [75]. Formation of the lamellipodia is crucial for cell migration and is a common dynamic surface extension exploited by cancer cells to invade and metastasize to secondary sites [76,77].

Moreover, PI3K/AKT can regulate Rho family members via crosstalk with other signaling molecules such as GSK-3β. On insulin activation, activated PI3K/AKT phosphorylates GSK-3β at serine 9, which leads to the inhibition of GSK-3β activity [78]. In gastric cancer cells, suppression of PI3K/AKT by PI3K inhibitor LY294002 or silencing AKT leads to decreased Wnt5-induced GSK-3β phosphorylation, which further causes a reduction in RhoA-dependent cell migration and actin remodeling [79]. Of note, inhibiting GSK-3β phosphorylation by pharmacological inhibitors restores RhoA activation and cell migration. Therefore, the suppression of GSK-3β could be another indirect mechanism for PI3K/AKT to regulate RhoA, and potentially the Rho family members, for remodeling of the actin cytoskeleton.

### 2.2. PI3K in Regulating Microtubules

Besides actin, microtubule is another critical cytoskeletal component that greatly influences the overall efficiency of directed cell migration [80,81,82]. Interestingly, the activation of PI3K/AKT signaling has been implicated in the regulation of microtubule stability, as well [83,84,85]. Within eukaryotic cells, the canonical microtubules are made up of numerous tubulin dimers (both α- and β-tubulin units) polymerizing into 13 linear protofilaments, which have been found to assemble around a hollow core to form a pseudo-helical structure [86,87]. Due to the alternative arrangement of the α- and β-tubulin units, microtubules are intrinsically polarized with the plus (+) end having the β-tubulin units exposed and the minus (−) end having the α-tubulin units exposed [88,89]. The work by Onishi et al. (2007) highlighted the importance of PI3K/AKT signaling in enhancing the stability of the microtubules [83]. In their experiments, they showed that the microtubules in fibroblast cells destabilized by either the addition of a pan-PI3K inhibitor LY294002 or the introduction of the dominant-negative form of AKT into the cells, which is in concordance with findings from earlier studies that showed enrichment of PI3K/AKT signaling and microtubule stability at the leading edge of the migrating cells [90,91]. The regulation of microtubule dynamics by the PI3K/AKT pathway can be transduced through GSK-3β. As discussed earlier, PI3K/AKT exerts an inhibitory effect on GSK-3β activity. The inhibition of PI3K/AKT signaling by LY294002 induces the activation of GSK-3β, which in turn, regulates a large repertoire of protein substrates that includes microtubule-associated proteins (MAPs) such as MAP-2, MAP-4, and tau [92]. The GSK-3β-mediated phosphorylation of these MAPs results in their diminished capacities to bind and stabilize the microtubules. As a result, the alteration to microtubule dynamics is thought to support cancer progression as it promotes uncontrolled motility in cancer cells [93]. Moreover, in the context of cancer biology, increasing studies show that cancer cells often contain modifications and mutations to the tubulin units that confer their resistance to conventional chemotherapy (microtubule-targeting agents) [64,94,95].

### 2.3. PI3K in Regulating Intermediate Filaments

The intermediate filaments, unlike the actin cytoskeleton and microtubules, comprise a large group of heterogeneous protein members, which are categorized into six major classes [96]. Another striking feature of the intermediate filaments, which sets them apart from the other cytoskeletal components, is that they do not control cell movements directly, but are important modulators of cell motility and often play crucial roles in providing mechanical support to the cells and tissues [97]. Often, the various classes of the intermediate filaments are also subjected to regulation by the PI3K/AKT signaling pathway. For example, keratin 18 (type I intermediate filament), along with its filamentous partner keratin 8 (type II intermediate filament), form the predominant pair of intermediate filament components in a typical epithelial cell, and their expression in cells is influenced by PI3K/AKT signaling, where overexpression of AKT increases keratin 8 and keratin 18 [98,99]. Although the activation of PI3K/AKT signaling increases the protein expression of both keratins, it is interesting to note that their number of mRNA transcripts remains relatively unchanged, suggesting that the activation of the PI3K/AKT signaling pathway promotes mRNA stability rather than the abundance of the transcripts. PI3K/AKT signaling is one of the most commonly dysregulated pathways in cancer [100]. Therefore, it is unsurprising that both keratin 18 and keratin 8 are often upregulated in most human cancer types [101,102]. Moreover, earlier studies also showed that an increase in the protein abundance of both keratin 18 and keratin 8 in cancer cells enhances the migratory and invasive capacities of cancer cells and alters their interactions with extracellular environments [103,104]. Similarly, the activation of PI3K/AKT signaling is also known to phosphorylate vimentin (type III intermediate filament). After phosphorylation by AKT, vimentin undergoes structural alterations, which lead to changes to its interactions with numerous intracellular components and its stability; PI3K inhibitor LY294002 and AKT inhibitor A563 antagonize these changes by decreasing the phosphorylation and stability of vimentin [105,106]. Such changes to the vimentin dynamics often result in the enhanced capacity of cancer cells to migrate and invade due to their ability to stabilize focal adhesions and enhance the mechanical strength of malignant cells [107].

## 3. Cytoskeleton in Regulating PI3K Signaling

### 3.1. Actin Cytoskeleton and Its Regulators in Regulating PI3K Signaling

The role of PI3K signaling in regulating the dynamics of the actin cytoskeleton has been well documented; the RhoGTPase family members have been extensively implicated in the migration and invasion of cancer cells on PI3K activation (refer to previous section for details). Recently, evidence emerged to suggest that family members of RhoGTPase play vital roles in regulating PI3K signaling (Figure 4). Multiple Rho GTPase members, including RAC, CDC42, and RhoG, cooperate to activate PI3K [108]. There appear to be isoform-specific effects of RhoGTPase on regulating different isoforms of p110, the catalytic subunit of PI3K. While p110α seems to be regulated by RAC/CDC42 indirectly, p110β has been identified as a direct target of RAC/CDC42 [108,109,110]. Upon activation of the upstream GPCR receptor, Dock family RAC-GEF Dock180, a member of guanine nucleotide exchange factor RAC-GEF that mediates the RAC activity, and its adaptor Elmo1, mediate the activation of RAC1 and CDC42 GTPase. Active RAC1 and CDC42 directly bind to p110β via the RAS-binding domain (RBD), which in turn, leads to activation of p110β [109]. The spatial distribution of PI3K isoforms might be important for Rho GTPase-mediated activation. RAC1 mediates the translocation of p110β to the lipid raft via direct interaction on upstream GPCR activation, which subsequently, turns on the PI3K signaling cascade [110]. In contrast, p110α resides predominantly in the non-raft region of the plasma membrane under GPCR activation [110]. Besides interacting with p110β, CDC42 can activate PI3K by suppressing the expression levels of PTEN and interfering with membrane localization of PTEN to the cell leading edge [111,112,113], suggesting another direction for RhoGTPase members in PI3K signaling via inhibiting their suppressors. In addition, activated RhoA GTPase recruits PTEN to the posterior of migrating cells to form a complex, leading to localized activation of PTEN and the polarized distribution [113]. Actin filaments may play a role in feeding back to PI3K in Rac-mediated actin polymerization, where pharmacological inhibitors of actin polymerization or depolymerization lead to decreased polarized PIP_3_ production in cells with PI3K being activated [114,115]. With the ability to act upstream of and trigger one another, PI3K and RAC/CDC42 could cooperatively create a positive feedback loop that potentiates and sustains PI3K signaling, and sustains the levels of active RAC/CDC42 at desired cellular localization, such as at the leading edge of migrating cells [116,117]. This localized positive feedback would lead to increased cell motility and invasion, which could eventually contribute to cancer progression.

Besides RAC/CDC42 GTPase, other families of actin-binding proteins may regulate PI3K in cancer. Gelsolin, an actin severing and capping protein, interacts with PI3K, leading to PI3K/AKT activation, and subsequently, to cell migration and dissemination of gastric cancer cells [118]. Members of the myosin family, the actin motors, were also shown to regulate the downstream effector, AKT, of the PI3K cascade. Myosin IB positively regulates the activation of AKT in the nucleus by binding to PTEN and preventing its nuclear translocation [119]. Myosin IIA and its activating enzyme MLCK are required for phosphorylation of AKT following MEK inhibition in triple-negative breast cancer cells [120]. Similarly, depletion of MYO18B suppresses the phosphorylation and activation of PI3K, as well as its downstream effectors such as AKT and mTOR in hepatocellular carcinoma [121]. The roles of actin polymerization proteins, such as Arp2/3, remain poorly understood in PI3K regulation. Since polymerized actin could feedback to PI3K activity and polarized PIP_3_ production [114,115], it is plausible that actin polymerization factors, which conventionally act downstream of the PI3K-RAC/CDC42 pathway, could regulate PI3K to provide additional directions for the positive-feedback circuit to promote cancer cell migration.

### 3.2. Microtubule Cytoskeleton in Regulating PI3K Signaling

Similar to their actin counterpart, and despite being tightly regulated by the PI3K pathway, microtubules can also act as upstream regulators of the PI3K signaling cascade. Several studies have suggested that microtubules can directly regulate PI3K signaling at different levels. PTEN is tethered to endocytic vesicles via phosphatidylinositol 3-phosphate (PI(3)P), to distribute around the microtubule network [122]. Through vesicular tethering, PTEN distribution is dependent on microtubules, and thus PI3K signaling can be regulated at distinct intracellular locations. The dynamics of microtubules play an important role in AKT activity. A functional microtubule cytoskeleton is required for phosphorylation of AKT induced by stimulation of insulin-like growth factor 1 receptor (IGF-1R), which lies upstream of PI3K [123,124]. Insulin receptor substrate 2 (IRS2) localizes to the microtubules, leading to AKT phosphorylation and activation in a manner dependent on the functional microtubule network [123]. A portion of IRS2 shows co-localization with microtubules on the cell membrane when the microtubules are stabilized for visualization, suggesting that IRS2 might travel along microtubules to interact with activated receptors at the cell membrane, to recruit downstream effectors like AKT. Activated AKT is then localized to microtubules via dynactin p150, a microtubule motor-binding protein [125]. The binding of AKT to microtubules sustains the phosphorylation status of AKT and thus the activation of the signaling cascade. Disassembly of microtubules switches off PI3K/AKT signaling through dephosphorylation and inactivation of AKT [125]. Interestingly, acetylated microtubules, which take the stabilized form, bind to AKT, leading to suppression of AKT activation and phosphorylation. There is more association of AKT with acetylated microtubules in the cytoplasm when microtubules’ acetylation levels are increased by treatment with α-tubulin acetyltransferase 1 inducers or deacetylase inhibitors. AKT may thus be sequestered inside the cytoplasm and prevented from being activated by PI3K on the plasma membrane under this condition [126]. Taken together, microtubules regulate PI3K/AKT signaling by directly participating in receptor substrate recruitment, sustaining activated AKT and sequestration of inactive AKT, and spatially regulating antagonists such as PTEN.

### 3.3. Intermediate Filaments in Regulating PI3K Signaling

The intermediate filaments, unlike the actin and microtubule counterparts, consist of a large group of members with a diversity of functions. Several class members of intermediate filaments have been shown to regulate PI3K signaling. Within the keratin family, several members can regulate PI3K/AKT signaling to modulate cancer cell behavior. Keratin 17 (K17) [127,128,129,130] and K80 [131] promote PI3K/AKT signaling via increased AKT phosphorylation, while K19 [132] plays inhibitory roles in PI3K/AKT signaling, as evidenced by the hyperactivation of AKT phosphorylation on the loss of this keratin. K8/18 is reported to both promote and inhibit AKT, depending on the cancer cell types and genetic background [133,134,135,136]. A possible mechanism for PI3K/AKT regulation is that keratin can act as a scaffold for signaling proteins to interact, when it physically binds with the adaptor protein 14-3-3. K17 and K18 bind to 14-3-3σ and retain the cytoplasmic localization of 14-3-3 under stimulation, which enables the recruitment of signaling molecules for PI3K/AKT activation [135,137]. Similarly, vimentin has been shown to interact with 14-3-3 [138]. Although poorly understood, vimentin may also feedback to PI3K/AKT signaling by interacting with 14-3-3, and affects the recruitment of signaling molecules through the 14-3-3 hub.

In addition to the cytoplasmic intermediate filament counterparts, lamins, which form the nuclear envelope, can also influence PI3K/AKT signaling by regulating the mRNA levels of PI3K subunits p110 and p85 [139]. Depleting lamin A/C abrogates PI3K activation and thus reduces the cell growth and migration/invasion of prostate cancer cells. Therefore, PI3K signaling, and the activities of intermediate filaments, coordinate in a co-regulatory manner, which impacts cancer cell behavior and disease progression, such as metastasis.

## 4. Clinical Relevance of PI3K/Akt-Cytoskeleton Crosstalk

### 4.1. Targeting PI3K/Akt in Cancer Treatment: PI3K Inhibitors in Clinical Trials

Dysregulation of PI3K/AKT signaling is a highly prevalent event in tumor progression [2,21,140]. Given its indispensable roles in tumor biology, it is unsurprising that efforts to develop novel therapeutic agents targeting this pathway have been unwavering for the past two decades (Table 1). Broadly speaking, PI3K inhibitors can be categorized into several classes; the more prominent ones include pan-PI3K inhibitors, isoform-selective PI3K inhibitors, and dual PI3K/mTOR inhibitors [141,142]. The pan-PI3K inhibitors can target all the isoforms of the catalytic p110 subunit (PI3Kα, PI3Kβ, PI3Kγ, and PI3Kδ) by blocking their ATP-binding sites [143,144].

An example of a well-studied pan-PI3K inhibitor would be buparlisib (NVP-BKM120), a reversible and orally bioavailable small molecule inhibitor [140,145]. Due to its potent anti-cancer properties in numerous preclinical studies, buparlisib has been propelled into clinical settings [146,147]. In the first-in-human study, the maximum tolerated dose (MTD) for buparlisib was found to be 100 mg/day [148]. In this phase-I dose-escalation trial, buparlisib showed some clinical activity in patients with advanced solid cancer types, though a multitude of adverse events such as asthenia, rash, hyperglycemia, decreased appetite, diarrhea, and nausea were observed. In a phase-II, open-label, single-arm study, similar adverse events were observed in patients who received buparlisib [149]. However, buparlisib was found to confer limited clinical benefits to patients despite the preselection of patients with tumors that harbor PI3K pathway alterations. Unfortunately, other clinical trials involving buparlisib in combination with other anti-cancer agents also failed to show significant clinical activity for patients suffering from advanced solid tumors [150,151,152].

Another prominent class of PI3K inhibitors is the dual PI3K/mTOR inhibitors, which have been regarded as a highly promising drug class due to the sheer importance of the PI3K/AKT/mTOR signaling axis in cancer biology [141,153]. Moreover, the potential to develop novel compounds acting as a single agent against two crucial targets within the same signaling axis was made possible due to the structural similarities between mTOR and the catalytic p110 subunit [154]. To date, there are only a few dual PI3K/mTOR inhibitors that remain relevant in clinical development, of which gedatolisib is one such candidate that has been evaluated against both solid and liquid tumors in the clinical setting [155,156]. In the first-in-human study, the MTD for gedatolisib was established to be approximately 154 mg/week [157]. Although adverse events such as mucosal inflammation, nausea, and hyperglycemia were commonly observed in patients, it was notable that antitumor activity was observed within the study—with two patients having partial responses and eight patients having lost-lasting stable disease. In a recent phase-I dose-escalation study involving gedatolisib in combination with carboplatin and paclitaxel for the treatment of patients with advanced solid tumors, 65 percent of the patients were found to achieve an objective response and 17 percent of the patients had stable disease [158].

Besides the traditional pan-PI3K and dual PI3K/mTOR inhibitors, another newer class of PI3K inhibitors, known as isoform-selective PI3K inhibitors, has also been starting to gain traction over the past few years as there is increasing evidence that supports the notion that different PI3K isoforms serve non-overlapping functions in cancer cells in a context-dependent fashion [159]. Moreover, these newer classes of PI3K inhibitors typically exhibit better safety profiles as they target only the relevant PI3K isoforms, thereby limiting the toxicities associated with pan-inhibition [2]. One of the most successful examples of isoform-selective PI3K inhibitors would be idelalisib (CAL-101), the first-in-class PI3Kδ-selective inhibitor, which received FDA approval for cancer patients with small lymphocytic lymphoma and follicular lymphoma in 2014 [160,161]. In a phase-I study involving 64 patients suffering from indolent lymphoma, idelalisib was administered to the patients at doses ranging from 50 to 350 mg (either once or twice each day) [162]. Although adverse events were reported in approximately 20 percent of the subjects, it was notable that under the idelalisib treatment regimens, 85 percent of the patients achieved disease regression. In another phase-III study, which evaluated the combinatory treatment regimen of idelalisib with rituximab (a monoclonal antibody that targets the transmembrane protein CD20 on B cells) for the treatment of relapsed chronic lymphocytic leukemia, patients who were given the combination treatment had better survival outcomes compared to patients who received rituximab as the sole therapeutic agent [163]. The median overall survival for patients in the combinatory treatment arm was reported to be 40.6 months, compared to 34.6 months for patients in the rituximab-only arm.

Another approach to target the PI3K/AKT signaling cascade in cancer treatment is to utilize AKT inhibitors, either mono-agent or in combinational therapy. To date, the two main classes of AKT inhibitors, the allosteric inhibitors and ATP-competitive inhibitors, have produced promising results in clinical development [164]. ATP-competitive inhibitors directly target the conserved ATP-binding pockets with high potency [165]. Although ATP-competitive inhibitors lack selectivity in general, many compounds have been identified with high binding potency and are undergoing extensive clinical evaluation among all AKT inhibitors. Capivasertib (AZD5363) and ipatasertib (GDC-0068) are promising candidates in this class, showing comparatively tolerable side effects in phase-I trials [166,167]. Common side effects include diarrhea, nausea, headache, hypertension, hyperglycemia, and fatigue [168]. Although monotherapies are observed to be ineffective in managing cancer progression [167], phase-II studies of these two compounds showed promising results in combinational therapy to combat multiple types of advanced or metastatic cancer, including ER+/HER2- breast cancer [169], triple-negative breast cancer [170], gastric cancer [171], and prostate cancer [172]. In these studies, addition of ATP-competitive inhibitors to clinically proven chemotherapeutic agents and hormone therapies, such as paclitaxel, fulvestrant, and abiraterone, showed improved treatment efficacy with prolonged disease-free, and sometimes overall, survival in patients. Phase-III clinical trials are underway to evaluate capivasertib and ipatasertib in combinational treatment for advanced breast and prostate cancers [173,174]. Pilot reports demonstrate that ipatasertib, in combination with atiraterone, improves progression-free survival for a subset of patients with metastatic castration-resistant prostate cancers who carry PTEN-loss, but no apparent improvement is observed in patients without such mutations [173]. Therefore, the genetic background of patients and tumors should be considered in the evaluation of treatment efficacies in ongoing trials with ATP-competitive AKT inhibitors.

Like ATP-competitive inhibitors, allosteric inhibitors have drawn attention for clinical development. Allosteric inhibitors have been shown to have good selectivity toward AKT rather than other kinases, due to the mechanism of maintaining AKT in its inactive conformation [175]. As a newer generation of ATP inhibitors, many allosteric inhibitors are in early-phase clinical trials [174,176]. MK-2206 in this class has been recently examined in phase-II clinical trials in combination with the aromatase inhibitor, anasterozole, to treat PIK3CA-mutant ER-positive and HER2-negative breast cancer, with no apparent improvement observed with the combinational treatment [177]. In comparison, HR-/HER2+ breast cancer patients treated with MK-2206 and neoadjuvant treatment had higher pathological complete response rates compared to patients receiving standard neoadjuvant therapy alone [178]. As the results for both ATP-competitive and allosteric inhibitors in clinical studies differ in patients with cancers of different genetic backgrounds, further studies are warranted to unravel the predictive biomarkers to maximize the therapeutic efficacies, to develop a precision treatment that utilizes AKT inhibitors.

**Table 1 cancers-14-01652-t001:** PI3K/AKT inhibitors in clinical development.

Drug Name	Phase	Treatment Composition	Disease Studied
Pan-PI3K inhibitors			
Buparlisib	I/II	Buparlisib monotherapy [148,149] Buparlisib + mFOLFOX6 [150] Buparlisib + abiraterone acetate [151] Buparlisib + enzalutamide [152]	Advanced solid tumors [148,150] Patients with solid or hematologic malignancies with PI3K pathway activation [149] Castration-resistant prostate cancer [151] Metastatic castration-resistant prostate cancer [152]
Dual PI3K/mTOR inhibitors			
Gedatolisib	I/II	Gedatolisib monotherapy [157,158]	Advanced solid tumors [157] Advanced solid tumors treated with palliative chemotherapy [158]
Isoform-selective PI3K inhibitors			
Idelalisib	III/FDA approved (for treating SLL)	Idelalisib monotherapy [161,162] Idelalisib + rituximab [163]	Relapsed indolent lymphoma [161,162] Relapsed chronic lymphocytic leukemia [163]
ATP-competitive AKT inhibitors			
Capivasertib	I/II	Capivasertib + fulvestrant [168,169] Capivasertib + paclitaxel [170]	PTEN-mutant ER + metastatic breast cancer [168] Estrogen receptor + HER2- metastatic/advanced breast cancer with aromatase inhibitor resistance [169] Metastatic triple-negative breast cancer [170]
Ipatasertib	II/III	Ipatasertib + mFOLFOX6 [171] Ipatasertib + abiraterone [172] Ipatasertib + abiraterone and prednisolone [173]	Locally advanced/metastatic gastric and gastroesophageal junction cancer [171] PTEN metastatic prostate cancer [172] Metastatic castration-resistant prostate cancer [173]
Allosteric AKT Inhibitors			
BAY 1125976	I	BAY 1125976 monotherapy [176]	Advanced solid cancer [176]
MK-2206	II	MK-2206 + anastrozole [177] MK-2206 + standard neoadjuvant therapy [178]	Stage II/III ER+/HER2- breast cancer with PIK3CA mutation [177] HR-/HER2+ breast cancer [178]

### 4.2. Targeting Cytoskeleton in Cancer Treatment

Cytoskeleton-targeting agents have been used in clinical practice in cancer treatment for a long time, with the majority of the agents used belonging to microtubule disruptors and anti-mitotic agents [56,179]. The microtubule-targeting agents (MTAs) in use belong to two main classes, microtubule destabilizing agents and microtubule-stabilizing agents, with opposite roles to play in microtubule polymerization and dissociation [180]. Both classes of drugs disrupt the dynamics and function of the microtubule network in cells, thus exerting potent anti-miotic effects on cancer cells to induce cell death [42]. MTAs can induce cell death in non-dividing cells via different mechanisms, such as inhibition of oncogenic signaling, vesicular trafficking, angiogenesis, and cell invasion, and thus MTAs remain as promising candidates for new chemotherapy development (Table 2) [35,43].

Two sites on tubulin are frequently recognized and bound by destabilizing agents, which are the ‘vinca’ domain and the ‘colchicine’ domain. Vinca alkaloids, originally extracted from Catharanthus roseus and other vinca plants, are one of the most used classes of chemotherapeutic agents [181,182]. There are five vinca alkaloids in clinical practice, vincristine, vinblastine, vindesine, vinorelbine, and vinflunine, which may be employed as a regime in combination with other chemotherapeutic drugs to treat multiple cancers [183]. Other vinca-site inhibitors also show promising therapeutic properties. Eribulin binds to the same site on tubulin as vinca alkaloids. It has been shown to have potent anti-cancer properties in various types of cancer preclinically, and to reduce abnormal tumor-associated vasculature [184,185]. Eribulin has been approved for treating metastatic breast cancer and liposarcoma in recent years [186,187], and is undergoing clinical trials for treating triple-negative breast cancer, which lacks effective treatment options clinically [188,189]. Dolastatin 10 is another non-vinca alkaloid microtubule-destabilizing agent, which exerts potent anti-mitotic and anti-tumor effects in cancers including small-cell lung cancer, ovarian cancer, prostate cancer, and breast cancer [190,191]. However, phase-2 clinical trials carried out in ovarian and prostate cancer showed no significant efficacy of dolastatin 10 when used in doses with good tolerance [192,193]. Recent advances in antibody-drug conjugates (ADCs) enable the safe usage of dolastatin 10 and its derivatives in high doses by targeting cancer cells via cancer-specific antigens such as CD30. Monomethyl auristatin E (MMAE), a synthetic dolastatin 10 analog, has been conjugated to the anti-CD30 monoclonal antibody and tested in several advanced-stage tumors, with good therapeutic efficacies achieved. Glembatumumab vedotin, an MMAE ADC, is undergoing preclinical and phase-I/II testing for breast cancer, recurrent/ refractory osteosarcoma, and advanced melanoma [194,195,196,197], while another MMAE ADC named brentuximab vedotin has been approved to treat anaplastic large-cell lymphoma and refractory Hodgkin lymphoma [198,199,200,201].

Colchicine-site binders, including colchicine and its analogs, represent another class of MDA undergoing clinical trials. Combretastatin A-4 (CA-4) and its prodrugs show efficacy in treating multiple hematological cancers and solid tumors in preclinical setups including acute myeloid leukemia and thyroid cancer [202,203]. Fosbretabulin (CA-4 phosphate) showed moderate effectiveness in combinational therapy for ovarian cancer and anaplastic thyroid cancer in clinical trials, with toxicities such as ataxia and cardiovascular symptoms observed [204,205,206,207]. Other combretastatin prodrugs, OXi4503 (combretastatin A1 diphosphate) and ombrabulin/AVE8062, are also undergoing clinical development for the treatment of acute myeloid leukemia (AML) and ovarian cancer, respectively [204,208]. Other colchicine-site binders under clinical trials include lisavanbulin (BAL101553) and plinabulin. Plinabulin is a non-conventional colchicine binding-site inhibitor, with its affinity for β-tubulin to inhibit tubulin polymerization [209,210]. Plinabulin is shown to exert anti-cancer effects in patients with solid tumors [211], and protective effects against neutropenia for non-small-cell lung cancer patients undergoing docetaxel treatment [212]. A phase-III clinical trial on combinational treatment of plinabulin and docetaxel for EGFR wild-type non-small-cell lung cancer is in progress [204]. If the outcome is favorable, palibabulin can serve as an alternative agent used in combination treatment to reduce the side effects of first-line regimens with docetaxel in NSCLC patients. Lisavanbulin, a water-soluble prodrug of avanbulin, shows anti-cancer activity against diverse treatment-resistant cancer models, including those resistant to conventional MTAs, in preclinical setups [213,214,215]. A phase I/II study for lisavanbulin in combination with radiotherapy to treat advanced solid tumors is ongoing [216]. Given the anti-cancer property of lisavanbulin in treatment-resistant cancer cells, it may be a promising anti-cancer agent in targeting intractable cancers.

Taxanes are among the most used drugs belonging to microtubule-stabilizing agents, where they bind to the taxane sites on β-tubulin [217]. Paclitaxel- and docetaxel-based regimes have been used as first-line chemotherapy in treating various solid cancers such as breast, ovarian, and lung cancers since the 1990s [179,218,219]. Because of treatment resistance in cancer cells and induction of toxicity to patients, novel taxane derivatives have been synthesized and tested both preclinically and in clinical trials. Cabazitaxel, with greater penetration of the blood-brain barrier compared to early-generation taxanes, has been developed to treat metastatic castration-resistant prostate cancer, and exhibits lower toxicity at low-dose administration compared to docetaxel [220,221]. Another approach to increase the therapeutic profile is to conjugate taxanes to nanoparticles/molecules designed to target cancer cells. These particles may consist of fatty acids, albumin, poly-l-glutamate, and other molecules [222,223,224]. Compared to unbound forms, nanoparticle-bound taxanes exhibit better drug delivery, lower toxicity, and potentially higher therapeutic value. Nab-paclitaxel, which is formed by binding paclitaxel to albumin, has achieved improved clinical efficacy in treating breast cancer compared with previous taxanes, and retains effectiveness in treating advanced/metastatic breast cancer patients, even for those who developed resistance to previous chemotherapy regimens (including taxanes-based treatment) [225,226,227,228].

Another microtubule-stabilizing agent under clinical evaluation is epothilone, which binds to the taxane-binding site on microtubules [229,230]. Compared to taxanes, epothilones exhibit good water solubility and brain-penetrating capacity, demonstrating higher efficacy in killing a broad spectrum of cancer cells in preclinical testing [231,232,233,234]. Moreover, epothilones interact with β-tubulin with a higher affinity compared to taxanes and are less susceptible to the effects of the drug efflux protein P-glycoprotein, hence reducing the chances of resistance development in cancer cells [232,235]. With these preclinical characteristics, epothilone B and its derivatives show promising treatment efficacy with tolerable side-effects and are in phase 2/3 trials [236,237]. Of note, ixabepilone, a semi-synthetic analog of epothilone, has been efficacious in treating breast cancer patients resistant to previous chemotherapy (including taxanes), with prolonged progression-free survival of patients [235,237,238]. This treatment efficacy was observed when ixabepilone was employed as a monotherapy in treating patients with metastatic/advanced treatment-resistant breast cancers. Ixabepilone synergizes in combination with capecitabine (which inhibits DNA synthesis) to prolong the overall survival of breast cancer patients who are resistant to anthracyclines and taxane-based regimes [237,238,239,240]. With the distinct characteristics of reduced resistance development and greater accessibility in the body, more epothilone members can be considered for evaluating their clinical values in cancer management.

**Table 2 cancers-14-01652-t002:** Microtubule-targeting agents in clinical development.

Drug	Phase	Treatment Composition	Disease Studied
Vinca-site binders			
Eribulin	II/ FDA approved (for metastatic breast cancer and liposarcoma)	Eribulin versus dacarbazine [186] Eribulin versus capecitabine [187] Eribulin + pembrolizumab [188,189]	Advanced liposarcoma or leiomyosarcoma [186] Advanced/metastatic breast cancer with prior anthracycline- and taxane-based treatment [187] HR+ HER2- metastatic breast cancer [188] Metastatic triple-negative breast cancer [189]
Glembatumumab vedotin (MMAE ADC)	II	Glembatumumab vedotin monotherapy [194,195,196,197]	Recurrent osteosarcoma [195] Advanced melanoma [196] Advanced glycoprotein NMB-expressing breast cancer [194] Metastatic glycoprotein NMB-expressing triple-negative breast cancer [197]
Brentuximab vedotin (MMAE ADC)	FDA approved	Brentuximab vedotin monotherapy [198,199,201]	Hodgkin’s lymphoma [199,201] Systemic anaplastic large cell lymphoma [198]
Colchicine-site binders			
Fosbretabulin	II	Fosbretabulin + pazopanib [205] Fosbretabulin + bevacizumab [206] Fosbretabulin + paclitaxel/carboplatin [207]	Recurrent ovarian cancer [205,206] Anaplastic thyroid carcinoma [207]
Combretastatin A1 diphosphate	I	CA1P monotherapy [208]	Relapsed or refractory acute myeloid leukemia [208]
Plinabulin	III	Plinabulin + docetaxel	Metastatic non-small cell lung cancer (NCT02812667)
Lisavanbulin	I/II	Lisavanbulin monotherapy [216]	Advanced solid tumors [216]
Taxane-site binders			
Cabazitaxel	III	Cabazitaxel versus docetaxel [220,221]	Metastatic castration-resistant prostate cancer [220,221]
Nab-paclitaxel	II/III	Nab-paclitaxel monotherapy [227] Nab-paclitaxel versus paclitaxel [226] Atezolizumab + nab-paclitaxel [228]	Advanced triple-negative breast cancer [228] Metastatic breast cancer patients with visceral metastases [227] Metastatic breast cancer [226]
Ixabepilone	III	Ixabepilone + capecitabine [238,240]	Metastatic breast cancer previously treated with anthracycline and taxanes [238,240]

### 4.3. Potential Crosstalk of PI3K Inhibitors and Cytoskeletal Disruptors in Clinical Treatment of Cancer

As discussed in the earlier sections, PI3K/AKT signaling regulates virtually all major classes of the eukaryotic cytoskeletal components, and hence, it is little wonder that many of these PI3K inhibitors exert great influence on the cytoskeletal dynamics, as well. For example, idelalisib is known to influence the distribution of chronic lymphocytic leukemia cells in patients at the cellular level by attenuating their migratory and invasive capacities [241]. Several preclinical studies have also shown that idelalisib can block these leukemia cells from migrating by both blocking chemotaxis directly and downregulating the production of the respective chemical stimuli in the stroma [242,243]. In the context of solid tumors, idelalisib has also been known to downregulate the expression of type-III intermediate filament vimentin and other key EMT markers by blocking the PI3K/AKT/GSK3β signaling axis in liver cancer cell lines [244]. As for buparlisib, it has been shown to destabilize the microtubule structures in glioblastoma cells, thereby decreasing their migratory and invasive properties [145,147]. However, based on our current understanding of the mechanistic actions of buparlisib, it has been suggested that the microtubule-destabilizing properties of buparlisib might be independent of its PI3K-inihibiting abilities [145,245].

Given the extensive crosstalk and feedback regulation between PI3K signaling and multiple cytoskeletal elements, it is reasonable to propose that PI3K inhibitors and cytoskeletal-targeting agents could be applied together to potentiate therapeutic efficacies. One important rationale for combining PI3K inhibitors and MTAs in treating cancer comes from studies on mitosis. MTAs are known to inhibit cancer cell proliferation by disrupting mitotic spindles, as microtubules and their associated proteins constitute many fundamental structures for mitotic spindles [179,246]. PI3K/AKT has also been shown to be indispensable for mitotic spindle formation, where p110α is activated on mitosis initiation to produce PIP_3_ at the midcortex in metaphase cells, for correct orientation of the spindles [247,248]. Due to the critical roles of PI3K/AKT signaling, inhibiting either PI3K or AKT through pharmacological inhibitors, such as LY294002 and MK-2206, leads to abnormalities in centrosome and mitotic spindle formation [247,249,250]. Of note, the class-II PI3K family member, PI3K-C2α, interacts with the transforming acidic coiled-coil containing protein 3 (TACC3) and clathrin heavy chain (CHC) complex to stabilize the kinetochore-microtubule for spindle formation [251]. These studies highlight the crucial roles of PI3K family members in regulating mitosis. One could reason that combining PI3K/AKT inhibitors and MTAs could potentiate the anti-mitotic activity of the treatment. Indeed, loss of PI3K-C2α further enhances the anti-mitotic effect of paclitaxel in a preclinical setup, where mitotic spindle disruption, abnormal metaphases, and cell death induction are observed to be increased with the combination of PI3K-C2α loss and paclitaxel treatment [251]. Moving into clinical development, phase-II clinical trials for HER2+ primary breast cancers have been conducted to evaluate the addition of buparlisib to paclitaxel and trastuzumab (a monoclonal antibody that targets HER2) treatment [252]. The initial results were not favorable, with early suspension due to toxicity concerns. However, a subset of patients with ER+ and HER2+ cancers showed better responses with the addition of buparlisib to paclitaxel and trastuzumab, compared to control groups receiving a placebo plus taxane-trastuzumab-based therapy. Therefore, future trials could consider combining newer generations of PI3K inhibitors and microtubule-targeting agents (MTAs) with enhanced safety profiles. These might include combinations of newer generations of MTAs, such as MDAs (e.g., MMAE ADCs), MSAs (e.g., ixabepilone), and second-generation PI3K inhibitors (e.g., taselisib and alpelisib). Moreover, customized selection of patients for treatment using tumor profiles and/or pathway-specific biomarkers may enhance the responses, as seen in the ER+/HER2+ subgroup of breast cancer patients. A recent phase-II study on squamous cell carcinoma of the head and neck examined the use of biomarkers to select patients who might respond to the combination of buparlisib and paclitaxel [253]. It was observed that patients with specific biomarker profiles, such as TP53 alterations or an HPV-negative status, had improved progression-free survival when treated with buparlisib and paclitaxel. Considering that MTAs are widely used in treating many solid cancers, and given the heterogeneous biology of cancers, future studies could aim to identify biomarkers and combinatorial regimes for stratified treatment with PI3K inhibitors and MTAs in different cancers at different stages.

## 5. Conclusions and Future Perspectives

Both the PI3K/Akt pathway and cytoskeleton are interlinked, acting as regulators and effectors for each other. The interplay and coordinated regulation of these two reshape cancer cell behavior, ultimately contributing to multiple aspects of disease progression including metastasis. Despite drug development to target PI3K and the cytoskeleton separately, targeting PI3K/AKT and the cytoskeleton together using combined therapy approaches should be considered for future drug development. Of note, drug resistance is a common obstacle to effective disease management when using the conventional MTA-based therapies, with PI3K/AKT as an important mechanism for drug resistance development [254]. Therefore, understanding the PI3K-cytoskeleton interplay will not only help to identify new druggable targets for future drug development but also provide insight into improving the current MTA-based chemotherapy. Future studies should continue to explore the combination approach of targeting PI3K-cytoskeleton elements, to develop effective customized and personalized treatments with improved safety profiles, as well as identify biomarkers for selecting patients who might reap the maximum benefit.

## Figures and Tables

**Figure 1 cancers-14-01652-f001:**
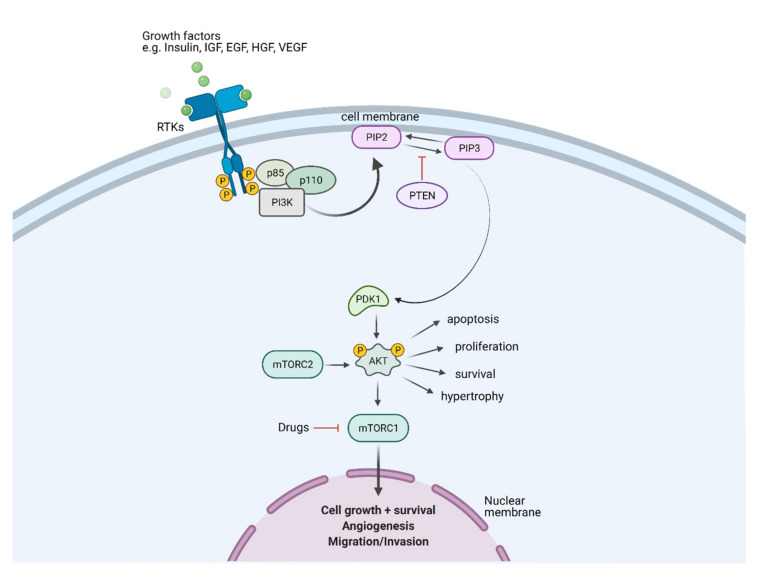
Schematic diagram illustrating the PI3K/AKT/PTEN/mTOR signaling pathway in a cell. A ligand-engaged RTK binds PI3K either directly or indirectly, removing the inhibitory action of its p85 subunit on the catalytic p110 subunit. In physiological conditions, growth factors stimulate PI3K, which subsequently phosphorylates the phospholipid substrate PIP_2_ to generate the second messenger PIP_3_. PIP_3_ recruits and activates several functional targets, such as AKT isoform, PDK1, and others. The lipid phosphatase PTEN converts PIP_3_ to PIP_2_, which terminates accentuation of the growth signal to maintain normal cellular and tissue homeostasis. RTKs, receptor tyrosine kinases; IGF, insulin-like growth factors; EGF, epidermal growth factor; HGF, hepatocyte growth factor; VEGF, vascular endothelial growth factor; PTEN, phosphatase and tensin homolog; PI3K, phosphoinositide 3-kinase; PIP_2_, phosphatidylinositol 4,5-bisphosphate; PIP_3_, phosphatidylinositol 3,4,5-trisphosphate; PDK1, phosphoinositide-dependent kinase-1; AKT, protein kinase B; mTORC1, mammalian target of rapamycin complex 1. The figure was created with BioRender.com (accessed on 14 February 2022) and was exported under a paid subscription.

**Figure 2 cancers-14-01652-f002:**
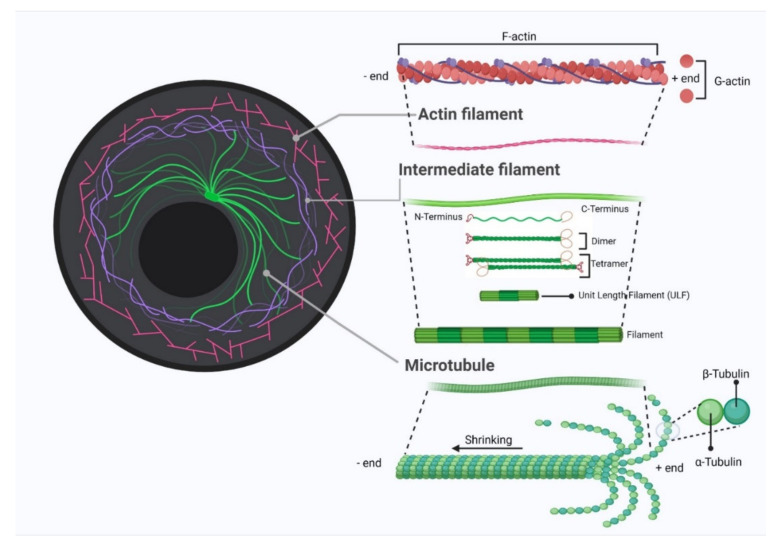
Schematic diagram of cytoskeleton structure that illustrates the three key components: actin filament, microtubule, and intermediate filament. The figure was created with BioRender.com (accessed on 14 February 2022) and was exported under a paid subscription.

**Figure 3 cancers-14-01652-f003:**
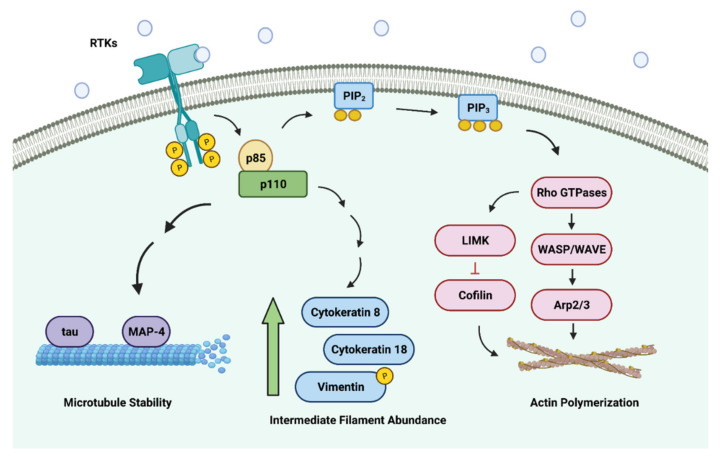
PI3K/AKT signaling axis regulates all major classes of cytoskeletal components. The activation of the PI3K/AKT signaling pathway plays key roles in the assembly of nascent actin filaments, polymerization of microtubules, and abundance of intermediate filaments. Collectively, these changes enhance the motility and migratory potential of cancer cells. Image created with BioRender.com (accessed on 14 February 2022).

**Figure 4 cancers-14-01652-f004:**
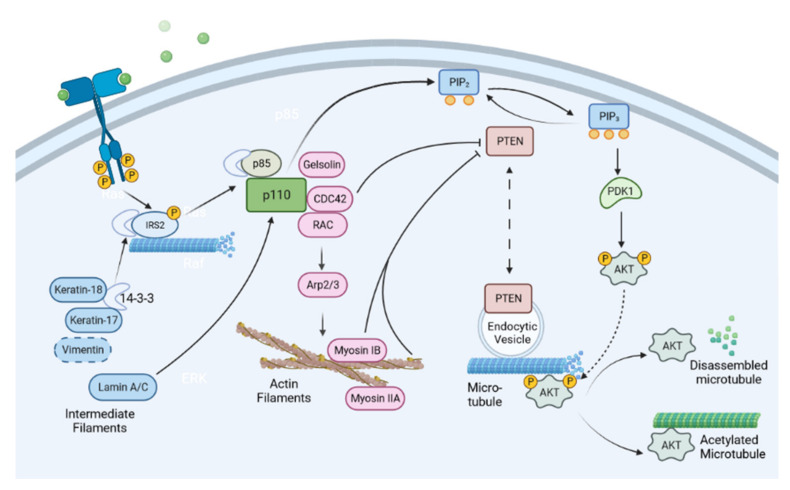
Cytoskeletal elements regulate PI3K/AKT signaling cascade. The three types of cytoskeleton collectively control the PI3K/AKT pathway. The cytoskeleton and its associated proteins play critical roles in regulating multiple steps and players of the PI3K/AKT signaling pathway, including signaling molecule recruitment, antagonist sequestration, and regulation of gene expression. Image created with BioRender.com (accessed on 14 February 2022).

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
