# Peer review of "PI3K/AKT Signaling Tips the Balance of Cytoskeletal Forces for Cancer Progression"

_cancers, 2022, doi:10.3390/cancers14071652_

Round 1
Reviewer 1 Report
The article entitled “PI3K/AKT signaling Tips the Balance of Cyto-skeletal Forces for Cancer Progression” is a comprehensive review work by authors. The manuscript discusses PI3K/AKT signaling, cytoskeletal remodeling, and the clinical approaches to that. I think the manuscript is very well written and highlights many known vital points to be considered. However, there are some points, which needs to be addressed or corrected.
- In the introduction, author mentioned three classes of PI3K, however, only one class is discussed. It would be informative, if authors can discuss the other classes very briefly.
- There are many published articles discusses the PAM pathway and its relationship with cytoskeletal remodeling leading to metastasis. So, this reviewer think to add rationale section in the manuscript and if is a update to the previously published knowledge, then I would recommend to mention that in the rationale line.
- Please correct grammatical error in sentence from 88-89.
- Is there any established link by your group or others linking PI3K/AKT/GSK-3B/B-Catenin? If it is so, does it have any correlation with cytoskeletal remodeling and metastasis.
- Please correct the reference as some of the references do not have doi or have weblinks in doi, and one reference has a web address in the literature title.
Author Response
The article entitled “PI3K/AKT signaling Tips the Balance of Cyto-skeletal Forces for Cancer Progression” is a comprehensive review work by authors. The manuscript discusses PI3K/AKT signaling, cytoskeletal remodeling, and the clinical approaches to that. I think the manuscript is very well written and highlights many known vital points to be considered. However, there are some points, which needs to be addressed or corrected.
- In the introduction, author mentioned three classes of PI3K, however, only one class is discussed. It would be informative if authors can discuss the other classes very briefly.
Response: We thank the reviewer for the suggestion. We have now included a brief description on all three classes of PI3K, lines 54-59 on page 3 in the revised manuscript
- There are many published articles discusses the PAM pathway and its relationship with cytoskeletal remodeling leading to metastasis. So, this reviewer thinks to add rationale section in the manuscript and if is a update to the previously published knowledge, then I would recommend to mention that in the rationale line.
Response: We thank the reviewer for pointing out this. We have revised the abstract and simple summary of the manuscript, to give a detailed rational and scope of the review. We added a section in the abstract as ‘we first discuss existing evidence in depth and then built on recent advances to propose new directions for therapeutic intervention. (page 2, lines 44-45).
- Please correct grammatical error in sentence from 88-89.
Response: We have revised this sentence in the manuscript into ‘These mutations, occurring at different components of PI3K signaling, can synergize or act independently to induce constitutive activation of PI3K and downstream effectors, which further induces cellular changes to enhance tumorigenesis and aggressiveness of established cancer’, from lines 99-102 on page 3.
- Is there any established link by your group or others linking PI3K/AKT/GSK-3B/B-Catenin? If it is so, does it have any correlation with cytoskeletal remodeling and metastasis.
Response: We thank the reviewer for suggesting looking into the link of PI3K/AKT with GSK3B in cytoskeleton remodeling. We have now included a study done by Liu et al., which studied the axis of PI3K/AKT/GSK-3β in cell migration. This studied showed that inhibiting PI3K by LY294002 or silencing AKT leads to decreased GSK-3β phosphorylation, which caused reduction in RhoA-dependent cell migration and actin remodeling. We have discussed the link of PI3K/AKT and GSK3B in regulating both actin cytoskeleton and microtubule dynamics in section 2.1 and 2.2 (lines 197-206 on page 6 and lines 231-234 on page 7), providing more mechanisms on how PI3K/AKT signaling can regulate cytoskeleton.
- Please correct the reference as some of the references do not have doi or have weblinks in doi, and one reference has a web address in the literature title.
Response: Thank you for pointing out this oversight. We have corrected the format of the references accordingly.
Reviewer 2 Report
This was a clear and concise review of the preclinical developments and investigation of the interplay between PI3K and cytoskeleton with potential therapeutic implications. An adequate overview of the current molecular and treatment landscape is provided prior to diving into the review. I do have several suggestions to improve the manuscript:
- Please highlight the significance of this topic in Simple Summary, which is not only abstract.
- As the authors pointed out, there are some preclinical and clinical applications of the use of PI3K inhibitors and MTAs. It’ll be more clear for the readers if the authors could list the clinical trials in a table.
Minor:
- Blue color word (Line 74)
- Extra number (line 1103)
- Format of DOI number in reference (line 739, 741, 751, 762, 784, and 1002)
Author Response
The review by Deng et al explores the interplay between PI3K/AKT signaling pathway and the cytoskeleton in cancer cells. Whether several reviews on PI3K/AKT are found in the literature, only very few have focused on this aspect. Hence the review is interesting and original. Overall it is clear and well written. Here are some points that should be addressed before acceptation for publication.
- Please highlight the significance of this topic in Simple Summary, which is not only abstract.
Response: We have revised the simple summary to highlight the significance of our work (line 23-27 on page 1). In addition, we have now also included a graphical abstract highlighting the importance of this topic.
- As the authors pointed out, there are some preclinical and clinical applications of the use of PI3K inhibitors and MTAs. It’ll be clearer for the readers if the authors could list the clinical trials in a table.
Response: We thank the reviewer for the suggestion. We have now listed the PI3K/AKT inhibitors and microtubule targeting agents in clinical trials in table 1 and 2 respectively.
- Minor comments: 1. Blue color word (Line 74); 2. Extra number (line 1103); 3. Format of DOI number in reference (line 739, 741, 751, 762, 784, and 1002)
Response: We thank the reviewer for pointing out this oversight. We have changed those errors accordingly.
Reviewer 3 Report
The review by Deng et al explores the interplay between PI3K/AKT signaling pathway and the cytoskeleton in cancer cells. Whether several reviews on PI3K/AKT are found in the literature, only very few have focused on this aspect. Hence the review is interesting and original. Overall it is clear and well written. Here are some points that should be addressed before acceptation for publication
- The authors generally speak about PI3K/AKT as a unity, however most of the effects rely on PI3K. It is therefore important to clearly delineate whether PI3K or AKT are at play. In addition the biological or chemical approach used to inhibit PI3K or AKT in the different quoted studies should be provided. (Name of inhibitors…)
- Related to point 1 the author did not describe any AKT inhibitor as if the review mainly focuses on PI3K
- In the last chapter of the review, the authors speculate on the relevance of PI3K/AKT cytoskeleton crosstalk in the clinic. Whereas the study clearly details the role of this pathway on actin filament, microtubules and intermediate filament, the role of PI3K/AKT on mitotic spindle is not discussed. Studies like Gulluni F et al cancer cell 2017 32 444-459 should be mentioned and the effect of PI3K/AKT on mitotic spindle should be described.
- In figure 1 mTORC2 needs to be added and the PIP2-PIP3 lipids should be at the cell membrane and not in the cytoplasm.
- The mechanisms (or any hypothesis) on how microtubules keep AKT phosphorylation need to be explained as usually AKT gets activated at the membrane.
Author Response
The review by Deng et al explores the interplay between PI3K/AKT signaling pathway and the cytoskeleton in cancer cells. Whether several reviews on PI3K/AKT are found in the literature, only very few have focused on this aspect. Hence the review is interesting and original. Overall, it is clear and well written. Here are some points that should be addressed before acceptation for publication.
- The authors generally speak about PI3K/AKT as a unity, however most of the effects rely on PI3K. It is therefore important to clearly delineate whether PI3K or AKT are at play. In addition the biological or chemical approach used to inhibit PI3K or AKT in the different quoted studies should be provided. (Name of inhibitors…)
Response: We have now included more details in the revised manuscript on component of the PI3K/AKT pathway that is the primary protein at play for the studies discussed. This additional information is mainly in section 2 on PI3K/AKT in regulating multiple aspects of cytoskeleton in cancer biology. For instance, we described that both PI3K and AKT are critical players for modulating RhoA-dependent actin remodeling and cell migration, as both PI3K inhibitor LY294002 or silencing AKT could reduce this process (lines 199-202 on page 6). We have also listed the approaches used to inhibit or activate PI3K or AKT in quoted studies in section 2 (pages 6-8).
- Related to point 1 the author did not describe any AKT inhibitor as if the review mainly focuses on PI3K
Response: We thank the reviewer for suggesting inclusion of AKT inhibitors into the manuscript. We have added a section in 4.1 to discuss AKT inhibitors in clinical trials (lines 441-480 on pages 12&13). We have also summarized the inhibitors in clinical phases in table 2.
- In the last chapter of the review, the authors speculate on the relevance of PI3K/AKT cytoskeleton crosstalk in the clinic. Whereas the study clearly details the role of this pathway on actin filament, microtubules and intermediate filament, the role of PI3K/AKT on mitotic spindle is not discussed. Studies like Gulluni F et al cancer cell 2017 32 444-459 should be mentioned and the effect of PI3K/AKT on mitotic spindle should be described.
Response: We thank the reviewer for pointing out the relevance of PI3K/AKT on mitotic spindle regulation to our manuscript. We have added a paragraph to discuss how PI3K/AKT and other classes of PI3K could regulate the mitotic spindle formation and mitosis. Moreover, we discussed the clinical context of PI3K/AKT’s roles in mitotic spindle regulation and discussed potential synergies of PI3K/AKT inhibitors and MTAs in clinical treatment via targeting mitotic spindle (lines 605-621 on page 17).
- In figure 1 mTORC2 needs to be added and the PIP2-PIP3 lipids should be at the cell membrane and not in the cytoplasm.
Response: We have revised the figure accordingly.
- The mechanisms (or any hypothesis) on how microtubules keep AKT phosphorylation need to be explained as usually AKT gets activated at the membrane.
Response: We proposed a mode on how AKT can be phosphorylated by microtubule. Based on the existing studies, insulin receptor substrate 2 (IRS2) might serve as the connection point between microtubule and AKT activation at the plasma membrane. IRS2 localizes to microtubule, thus it might be transported to plasma membrane along the microtubule. IRS2 may recruit AKT and downstream effectors to the site of activation. Once AKT is activated, microtubule would maintain its phosphorylated status. This is described in lines 339-342 (on page 10) in the revised manuscript.
Round 2
Reviewer 3 Report
one minor correction in table 1 ATP competitive AKT inhibitor and not ATP competitive ATP inhibitor